# Regulatory Control of Rishirilide(s) Biosynthesis in *Streptomyces bottropensis*

**DOI:** 10.3390/microorganisms9020374

**Published:** 2021-02-12

**Authors:** Olga Tsypik, Roman Makitrynskyy, Xiaohui Yan, Hans-Georg Koch, Thomas Paululat, Andreas Bechthold

**Affiliations:** 1Institute for Pharmaceutical Biology and Biotechnology, Albert-Ludwigs-Universität, 79104 Freiburg, Germany; olga.tsypik@pharmazie.uni-freiburg.de (O.T.); roman.makitrynskyy@pharmazie.uni-freiburg.de (R.M.); yanxh@tjutcm.edu.cn (X.Y.); 2Faculty of Medicine, Institute for Biochemistry and Molecular Biology, ZBMZ, Albert-Ludwigs-Universität, 79104 Freiburg, Germany; hans-georg.koch@biochemie.uni-freiburg.de; 3Department of Chemistry-Biology, Institute of Organic Chemistry II, Universität Siegen, 57068 Siegen, Germany; paululat@chemie.uni-siegen.de

**Keywords:** *Streptomyces*, secondary metabolites, type II polyketide, regulation, SARP, MarR, rishirilide

## Abstract

Streptomycetes are well-known producers of numerous bioactive secondary metabolites widely used in medicine, agriculture, and veterinary. Usually, their genomes encode 20–30 clusters for the biosynthesis of natural products. Generally, the onset and production of these compounds are tightly coordinated at multiple regulatory levels, including cluster-situated transcriptional factors. Rishirilides are biologically active type II polyketides produced by *Streptomyces bottropensis*. The complex regulation of rishirilides biosynthesis includes the interplay of four regulatory proteins encoded by the *rsl*-gene cluster: three SARP family regulators (RslR1-R3) and one MarR-type transcriptional factor (RslR4). In this work, employing gene deletion and overexpression experiments we revealed RslR1-R3 to be positive regulators of the biosynthetic pathway. Additionally, transcriptional analysis indicated that *rslR2* is regulated by RslR1 and RslR3. Furthermore, RslR3 directly activates the transcription of *rslR2*, which stems from binding of RslR3 to the *rslR2* promoter. Genetic and biochemical analyses demonstrated that RslR4 represses the transcription of the MFS transporter *rslT4* and of its own gene. Moreover, DNA-binding affinity of RslR4 is strictly controlled by specific interaction with rishirilides and some of their biosynthetic precursors. Altogether, our findings revealed the intricate regulatory network of teamworking cluster-situated regulators governing the biosynthesis of rishirilides and strain self-immunity.

## 1. Introduction

Streptomycetes are soil bacteria that are well-known for their complex morphological differentiation and ability to produce a plethora of natural products with pharmaceutical value [1,2,3]. To date, the genus *Streptomyces* produces more than half of the clinically used antibiotics and is still considered to be a promising source for novel antibiotics discovery [4].

To coordinate development and secondary metabolite production, *Streptomyces* developed a complex regulatory network with multiple overlapping cascade systems. This includes sigma factors, transduction systems, and transcriptional factors of different families [5,6,7].

The onset and level of natural product production are determined in accordance to environmental signals, nutrition, and the physiological state of growing cells [6,8]. In *Streptomyces*, specific low molecular weight signalling molecules such as gamma-butyrolactones and guanosine penta/tetraphosphate ((p)ppGpp) trigger secondary metabolism production and morphological differentiation [9,10,11,12]. Recently, intracellular second messenger cyclic di-GMP was also shown to exert a regulatory function on biosynthesis of natural products [13,14,15].

Genes involved in synthesis of natural product are clustered together. In general, *Streptomyces* chromosome harbors 20–30 biosynthetic gene clusters for secondary metabolites [6,16]. Typically, a biosynthetic gene cluster encodes a cluster-situated regulator, which directly controls the transcriptional levels of genes for production of the cognate natural product. Interestingly, some clusters can contain none or several regulatory genes [6,17,18,19,20]. The cluster-situated regulator in turn is controlled by higher-level regulatory elements like pleiotropic or global regulators [21]. Most of the cluster-situated regulators act as activators, and belong to the SARP (*Streptomyces* antibiotic regulatory proteins) family [6,21]. The most intensively studied SARPs are ActII-ORF4 and RedD, activators of actinorhodin and undecylprodigiosin production in *S. coelicolor*, respectively [6,22,23,24,25]. The promoter region of *actII-ORF4* is a regulatory target of few superior transcriptional factors including AdpA, AbsA2, DasR, AfsQ1 and others [26,27,28,29].

*Streptomyces bottropensis* is a known producer of the macrocyclic peptide antibiotic bottromycin and type II polyketides—rishirilides and mensacarcin [30]. Rishirilide A and B act as inhibitors of α2-macroglobulin, whereas rishirilide B exhibits an additional inhibition activity towards glutathione-S-transferase [31,32]. Genes for rishirilides biosynthesis (*rsl*-genes) are clustered together and were recently functionally characterized [33].

Rishirilides are synthesized from the valine-derived isobutyl-CoA starter unit and malonyl-CoA extenders, followed by decarboxylation of one acetate unit giving a methyl group. The minimal PKS complex derives a linear tricyclic intermediate, which is further modified by cluster-encoded tailoring enzymes [33,34]. Interestingly, the rishirilide scaffold is formed in two key steps: a reductive ring opening of an epoxide moiety on polycyclic intermediate by the flavin-dependent reductase RslO5, followed by an extraordinary oxidative carbon backbone rearrangement catalyzed by flavin-dependent Baeyer-Villiger monooxygenase RslO9 [35].

In addition to biosynthetic and transporter genes, four regulators are encoded by the *rsl*-gene cluster. However, detailed function of these transcriptional factors in the biosynthesis of rishirilides remained unclear. In this work, we have generated *rslR1*, *rslR2*, *rslR3*, and *rslR4* deficient mutants and analyzed their ability to produce rishirilides. Inactivation of *rslR1*-*R3* completely abolished biosynthesis, whereas a deletion of *rslR4* had no substantial effect on a production level. The overexpression of *rslR1*, *rslR2*, and *rslR3* in *S. bottropensis* greatly increased titers of rishirilides accumulation, suggesting that RslR1, RslR2, and RslR3 are positive regulators. In order to identify putative interconnections between studied regulators, the expression of regulatory genes was analyzed in wild type and in mutant strains by RT-PCR and a GusA reporter system. RslR3 was shown to exert positive control on the transcription of *rslR2*. This finding was further confirmed by direct binding of RslR3 to the *rslR2* promoter. Moreover, both RslR1 and RslR2 negatively control transcription of *rslR2*.

RslR4 is involved in the regulation of strain immunity. It modulates transcriptional levels of the MFS transporter *rslT4* and its own gene. We also demonstrated that the final biosynthetic products and key pathway intermediates interact with RslR4 and release the protein from a target DNA.

## 2. Materials and Methods

### 2.1. Bacterial Strains, Plasmids, and Growth Conditions

The bacterial strains and plasmids used in this work are described in Appendix A. *S. bottropensis* strains were grown at 28 °C in liquid YMPG medium [36] supplemented with 0.2% valine for rishirilide production (YMPGv), and on MS agar for harvesting spores and plating conjugations. *Escherichia coli* strains were grown in LB medium at 37 °C unless otherwise stated. Where necessary, the media were supplemented with antibiotics: apramycin (50 µg mL^−1^), kanamycin (50 µg mL^−1^), spectinomycin (100 µg mL^−1^), chloramphenicol (25 µg mL^−1^), and ampicillin (100 µg mL^−1^). All antibiotics were purchased from Carl Roth, Karlsruhe, Germany.

### 2.2. Generation of Gene Deletion Mutants

The double cross-over mutants were generated using Red-Et mediated technology [37]. Initially, an integrase gene (*int*) on a cosmid cos4, carrying the *rsl*-gene cluster, was inactivated to prevent an integration of the final construct into *S. bottropensis* chromosome. For this purpose, cos4 was introduced into *E. coli* BW25113 (pIJ790) where the integrase gene was replaced by an ampicillin resistance gene, which was amplified from pBluescript using primer pair Int-aatP::amp-f/r with *int*-homology extensions. The resulting cosmid cos4-int::bla was confirmed by PCR analysis and transformed into *E. coli* BW25113 (pIJ790).

Next, the spectinomycin resistance cassette was amplified from pCDFduet using primer pair rslR-RedET-Fw and rslR-RedET-Rv listed in Appendix A. The obtained corresponding PCR products with homology extensions to *rslR1*, *rslR2*, *rslR3*, and *rslR4* were individually electroporated into *E. coli* BW25113 (pIJ790) cos4-int::bla to replace the desired regulatory gene by the spectinomycin resistance cassette. The resulting constructs cos4-int::bla-rslR1::Sp, cos4-int::bla-rslR2::Sp, cos4-int::bla-rslR3::Sp, and cos4-int::bla-rslR4::Sp were confirmed by PCR and individually introduced into *S. bottropensis* via intergeneric conjugation with *E. coli* ET12567 (pUZ8002), harboring the desired construct. The double cross-over mutants were screened for resistance to spectinomycin and sensitivity to apramycin, and further confirmed by PCR analysis.

### 2.3. Construction of the Plasmids for Complementation and Overexpression

For complementation experiment, a regulatory gene together with a 500 bp upstream promoter region was amplified by PCR using chromosomal DNA of *S. bottropensis* as a template and primer pair listed in Appendix A. The obtained PCR product was purified, digested with XbaI, and cloned into XbaI/EcoRV sites of the φC31-based integrative plasmid pSET152. The yielded plasmid pSET-rslR was conjugated into the respective gene deletion mutant. To complement *S. bottropensis* ∆R3, plasmid pTES-rslR3 was constructed. For this reason, a coding sequence of *rslR3*, along with the ribosome binding site (RBS), was amplified with R3-XbaI-pTESa/rslR3MuDet-Fw primers (Appendix A) and cloned into XbaI/EcoRV-cleaved pTES. In resulting pTES-rslR3, the expression of *rslR3* is under control of constitutive *ermE*-promoter.

To overexpress regulatory genes, their coding sequences, along with RBS, were amplified from *S. bottropensis* chromosome with primers listed in Appendix A. The obtained PCR fragments were digested with respective restriction endonucleases, which recognition sites are highlighted in the primer sequence, and then ligated into the same sites of the linearized multicopy expression vector pUWL-H. In these plasmids, the expression of genes is under control of the strong constitutive promoter *ermEp*.

### 2.4. Strain Fermentation and Analysis of Rishirilides Production

Spores of the wild type and of mutant strains were inoculated into 50 mL of YMPGv in 300-mL flasks and grown for five days with rotary shaking. Then, supernatant was separated from cell pellet by centrifugation and extracted with the same volume of ethyl acetate. After evaporation, the extracts were dissolved in methanol and used for UHPLC-MS analysis as described before [35]. To evaluate the production level, total rishirilide was quantified. Total rishirilide refers to rishirilide A, B, D, and lupinacidin A.

### 2.5. Total RNA Isolation and RT-PCR

Total RNA was isolated from *S. bottropensis* strains, grown for 72 h in YMPGv medium, using RNeasy Mini kit (Qiagen, Venlo, Netherlands) according to recommendations from the supplier. To get rid of DNA contaminations, RNA samples were treated with DNaseI (Promega, Madison, Wisconsin, United States) and checked by PCR prior cDNA synthesis. Then, cDNA was synthesized from 1 µg of total RNA using Protoscript II Reverse Transcriptase (NEB, Ipswich, MA, USA) according to instructions of the manufacturer. To analyze the transcription profile of genes of interest, 200 ng of cDNA were used as a template in PCR reaction with primers listed in Appendix A. Obtained PCR products were separated on a 1.5% agarose gel. Total RNA was isolated from three independent replicates.

### 2.6. Gus-Assay

Promoter regions of *rsl*-genes were amplified by PCR using *S. bottropensis* chromosomal DNA as a template and primer pairs listed in Appendix A. The PCR-products were purified, digested with XbaI and KpnI restriction endonucleases, and cloned into pGUS. Recombinant plasmids were conjugated into *S. bottropensis* and in the respective mutants. β-glucuronidase activity was measured in strains lysate after 72 h (*rslR1*p and *rslR2*p) and 48 h (*rslR4*p and *rslT4*p) of growth in YMPGv medium. Measurement of β-glucuronidase activity was carried out as described before [38] in biological triplicate.

### 2.7. Production and Purification of RslR3DBD

To produce native or truncated versions of RslR3, the coding sequences were amplified by PCR using primer pairs listed in Appendix A. The PCR products were digested with NdeI and HindIII and inserted into the respective sites of pET28a. The resulting plasmids were confirmed by sequencing and further introduced into *E. coli* BL21 (DE3) Star. RslR3DBD was produced with N-terminal hexahistidine tag. *E. coli* BL21 (DE3) Star pET28a-R3DBD was incubated in LB medium at 37 °C until the optical density at 600 nm reached 0.5, then IPTG (0.1 mM) was added and incubation continued at 18 °C overnight. The cells were harvested by centrifugation and resuspended in a binding buffer (50 mM Na_2_HPO_4_, 0.5 M NaCl, 2% glycerol, pH 7.5 supplemented with PMSF and DNaseI). After cell disruption by sonication, the cleared lysate was applied to Ni-NTA and washed with increasing concentration of imidazole (binding buffer containing 40, 80, and 100 mM imidazole). Protein was eluted with 250 mM imidazole and analyzed by SDS-PAGE.

### 2.8. Production and Purification of RslR4

The open reading frame of *rslR4* along with 51 bp located upstream of the gene was amplified by PCR using chromosomal DNA of *S. bottropensis* as a template and primer pair R4-NcoI-17/R4-XhoI (Appendix A, see Appendix A for RslR4 sequence). The PCR product was digested with NcoI and XhoI and ligated into the same sites of pET28a. The resulting construct pET28-rslR4 was sequenced and introduced into *E. coli* BL21 (DE3) Star. To produce RslR4, the culture was grown in LB medium until optical density reached 0.4. After the addition of 0.2 mM IPTG, the culture was grown for another four hours at 22 °C. Cells were harvested by centrifugation and directly used for protein purification or frozen at −20 °C until required. The cell pellet was dissolved in a buffer (50 mM TrisHCl, 0.5 M NaCl, 20 mM imidazole, pH 8.0) supplemented with PMSF and DNaseI, and disrupted by two passages through a French pressure cell press. Lysate was separated from insoluble fraction by centrifugation and applied for protein binding with Ni-NTA resin. After that, resin was washed with the buffer containing 40 mM imidazole and protein was eluted by a linear gradient from 100 to 200 mM imidazole. Eluted fractions were analyzed for the presence of RslR4 by SDS-PAGE. Samples with RslR4 were collected and protein concentration was estimated by Bradford assay.

### 2.9. Electrophoretic Mobility Shift Assay

Putative promoter regions of *rsl*-genes were amplified by PCR using *S. bottropensis* chromosome as a template and primers listed in Appendix A. The PCR products were labeled at 5′-end with [γ-^33^P]-ATP using T4 polynucleotide kinase. DNA-binding reactions were carried out at 28 °C for 15 min (or for 25 min with RslR3DBD) in a binding buffer (20 mM TrisHCl pH 7.9, 100 mM KCl, 10 mM MgCl_2_, 1 mM EDTA, 1 mM DTT, 12.5% glycerol) supplemented with 1 µg poly(dI-dC) and 4.5 µg BSA. DNA-protein complexes were resolved on 8% polyacrylamide native gels in TBE buffer and visualized by phosphorimaging on Typhoon FLA7000. For retardation assay, unlabeled DNA probe was added to the reaction mixture at 10- to 500- molar excess (as indicated in results section).

To test rishirilides and pathway intermediates as ligands of RslR4, compounds were incubated with the protein (100 nM) prior adding *rslR4-T4* intergenic region to the reaction. The following concentrations of the ligands were used: 1, 10, 50, 100 µM and 0.25, 0.5, 1 mM. The tested ligands were purified as described before [33,35] or in Appendix A (information about structure elucidation of compounds isolated from ∆*rslO7* mutant).

## 3. Results

### 3.1. Biosynthesis of Rishirilides Is Positively Regulated by RslR1, RslR2, and RslR3 Cluster-Situated Regulators

The *rsl*-gene cluster encodes four regulatory proteins. Analysis of domain architecture of RslR1 and RslR2 identified an N-terminal winged helix-turn-helix (HTH) DNA-binding and Bacterial Transcriptional Activation (BTA) domains that are typical for members of the SARP-family regulatory proteins. RslR3 is a 1112-amino acid transcriptional factor which belongs to large SARPs and has multi-domain architecture similar to AfsR and PolY [19,39]. Like RslR1 and RslR2, it consists of an N-terminal SARP domain, which is followed by an additional ATPase (AAA/NB-ARC) domain and a C-terminal tetratricopeptide repeat (TPR) domain that usually mediates protein–protein interactions. RslR4 is a member of the MarR-family (Multiple Antibiotic Resistance Regulator) transcriptional factors.

To investigate the role of these regulators in rishirilide biosynthesis, we inactivated them individually in the *S. bottropensis* chromosome by in frame replacement of their coding sequences with the spectinomycin resistance gene. To estimate rishirilides production, the resulted mutants together with wild type strain were grown in YMPGv medium for five days and their metabolite profiles were analyzed using UHPLC-MS. Inactivation of *rslR1*, *rslR2*, and *rslR3* completely abolished production of all rishirilide compounds indicating their positive role in the biosynthesis (Figure 1a). Inactivation of *rslR4* did not distinctly affect rishirilides production.

To exclude the possibility of any polar effect and to confirm that changes in the production of rishirilides were solely due to the inactivation of the targeted regulatory gene, the integrative pSET152-based construct carrying a native *rslR* gene allele together with its promoter region was introduced into mutants. Complementation of *rslR1* and *rslR2* mutants restored the production of rishirilides. Complementation of *S. bottropensis* ∆R3 failed when *rslR3* was expressed from its putative promoter region in pSET-rslR3. However, complementation with pTES-rslR3, where the regulatory gene is under control of the constitutive *ermE*-promoter (Figure 1b,c), restored the ability of *rslR3* mutant to produce rishirilides. This result suggests that *rslR3* lacks its own promoter, that was further confirmed by gene co-expression analysis (see Section 3.3). In order to overexpress *rslR1*-*R3* in the wild type strain, replicative multi-copy plasmids expressing each regulatory gene from the *ermE*-promoter (pUWLH-rslR1, pUWLH-rslR2, and pUWLH-rslR3) were constructed and conjugated into *S. bottropensis*. Indeed, overexpression of *rslR1*, *rslR2*, and *rslR3* in the wild type increased rishirilides production from 3- to 5-fold (Figure 1d). These results are in agreement with the gene deletion experiments, and indicate that the SARP regulatory proteins RslR1, RslR2, and RslR3 are positive regulators of the *rsl*-gene cluster. Moreover, the activity of all three proteins is required for rishirilide biosynthesis. On the other hand, RslR4 might be not involved in direct regulation of transcription of structural genes. However, it may control strain self-resistance to produced compound(s).

### 3.2. Expression Pattern of Regulatory Genes in Wild-Type and Mutant Strains

*S. bottropensis* initiates rishirilides production after 48 h of growth in fermentation medium. To determine putative interconnections between regulatory genes, their expression was analyzed in the wild type and in the *rslR1*-, *rslR2*-, *rslR3*-, and *rslR4*-disruption mutants. Total RNA was isolated from the strains after 72 h of growth and genes transcripts were analyzed by RT-PCR. As a control, the *hrdB* gene, which encodes a RNA polymerase principal sigma factor, was used.

In the semi-quantitative RT-PCR analysis, the level of transcripts of *rslR4* was the same in the wild type strain and in *S. bottropensis* ΔR1, *S. bottropensis* ΔR2, and *S. bottropensis* ΔR3 (Figure 2a). This result indicates that *rslR4* transcription is independent from a control by RslR1, RslR2 or RslR3. Transcriptional profiles of *rslR3* and *rslR1* were very similar in all analyzed strains as well, indicating that both genes are independently expressed. In contrast, the low level of *rslR2* transcripts in *S. bottropensis* ΔR3 indicates that *rslR2* may be positively regulated by RslR3. In addition, transcription of *rslR2* might be under a negative control of RslR1 due to the increased abundance of *rslR2* transcripts in *S. bottropensis* ∆R1 (Figure 2a).

To verify these results as well as to investigate the possibility of self-regulation, expression of *rslR1* and *rslR2* at transcriptional level was analyzed in the wild type and in *S. bottropensis* ΔR1, *S. bottropensis* ΔR2, and *S. bottropensis* ΔR3 using the GusA-reporter system [38]. Promoter regions of *rslR1* and *rslR2* were cloned upstream of the β-glucuronidase gene and the obtained plasmids were then conjugated into the wild type and mutant strains. Activity of GusA in strain lysates was measured after 72 h of growth. Strains carrying the empty vector pGUS were used as a control (Figure 2b). There was no significant difference in the *rslR1* promoter activity in the wild type and in *S. bottropensis* ΔR1, *S. bottropensis* ΔR2, and *S. bottropensis* ΔR3 (Figure 2c). These data are in agreement with the results of the RT-PCR experiments. In contrast, the transcription of *rslR2* was strongly downregulated in the *rslR3* mutant and increased two-fold in *S. bottropensis* ΔR1 and *S. bottropensis* ΔR2 (Figure 2d). These data indicate that RslR3 and RslR1 can act as positive and negative regulators, respectively. Moreover, RslR2 acts as a negative regulator of its own transcription.

In summary, these data suggest that transcription of *rslR2* is controlled by multiple regulators. It is activated by RslR3 and downregulated by RslR1 as well as by RslR2. Thus, RslR3 controls the pathway specific regulator RslR2. Finally, transcription of *rslR4* is independent from the other Rsl-regulatory proteins.

### 3.3. Transcriptional Organization of rsl-Genes

To identify putative promoter regions in the cluster, the transcriptional organization of the *rsl*-genes was determined. The co-transcribed genes were mapped by RT-PCR with primers, which anneal across gene pairs using cDNA from the wild type strain as a template. The obtained PCR products represent junction sequences between the genes (Figure 3). Amplification products between *rslK1* and *rslK2*, as well as *rslK3* and *rslA*, indicate that the minimal PKS genes form a single operon. The PCR products obtained for gene sets *rslT1*-*rslT2*, *rslT3*-*rslO1*, *rslO1*-*rslO2*, *rslO2*-*rslP*, and *rslP*-*rslR1* indicate that genes between *rslR1* and *rslT1* are transcribed together. The faint PCR products obtained for *rslT1*-*rslT2* and *rslP*-*rslR1* indicate that there may be promoters located upstream of *rslT1* and *rslP* in addition to a promoter upstream of *rslR1*. The PCR products were also obtained for further gene pairs (Figure 3) but not for *rslC2*-*rslO3*. These data indicate that genes between *rslR2* and *rslO4*, *rslR4* and *rslR3*, and *rslT4* and *rslH* clearly form transcription units and that there are promoters located upstream of *rslC2*, *rslO3*, *rslR2*, *rslR4*, and *rslT4*. The weak PCR fragment for *rslR3*-*rslR2* may suggest the co-transcription of both genes. However, successful complementation of *S. bottropensis* ∆R2 by pSET-rslR2 confirmed the existence of a promoter upstream of *rslR2* because in our complementation construct *rslR2* can only be expressed from its own promoter. Interestingly, the transcription unit *rslR4*-*rslO6*-*rslR3* indicates that *rslR3* lacks its own promoter region. This explains the ability to complement *S. bottropensis* ∆R3 only by expressing *rslR3* from the constitutive *ermE*-promoter. Complementation failed when a fragment containing *rslR3* and the region between *rslO6* and *rslR3* was used.

### 3.4. RslR3 is a Multi-Domain Protein That Regulates rslR2

To further confirm direct interaction of RslR3 with the promoter region of *rslR2*, a radio-labeled DNA probe was used in an electrophoretic mobility shift assay with a recombinant RslR3. Beside the *rslR2*p, *rslR1*p and promoters of *rsl*-transcriptional units were used. Production of the recombinant full-length RslR3 with an N-terminal hexa-histidine tag resulted in insoluble protein, which hampered the investigation of its DNA-binding ability. It has been shown that truncated versions of AfsR containing either the SARP and ATPase domains or only the SARP domain, show the same DNA interaction pattern as the full-length protein [39]. As domain organization of RslR3 is similar to AfsR, an analogous experiment was performed. Two truncated versions of RslR3, consisting of only the SARP-domain (RslR3DBD) and the SARP-domain accompanied by the ATPase domain (RslR3DBDP) were constructed. Only the RslR3DBD version could be produced as a soluble protein and was used for further studies. Increasing concentrations of purified RslR3DBD were incubated with a DNA fragment containing the promoter region of *rslR2* (Figure 4a). DNA-protein complexes were formed at concentrations of 0.5–1 µM. RslR3DBD yielded a bit smeared and broad bands, which could be due to instability of the truncated protein. To confirm a specific binding of RslR3DBD to *rslR2*p, the protein (1 µM) was incubated with labeled DNA probe at the presence of 10 to 500-fold excess of unlabeled DNA. Unbound labeled *rslR2*p was detectable indicating a specific binding of RslR3DBD to the promoter region of *rslR2* (Figure 4b). No binding was observed using *rslR1*p or any other tested promoter region (Figure 4c,d).

Altogether, these results are in a line with data of RT-PCR and in vivo reporter system experiments. This finding suggests that *rslR2* is directly activated by RslR3 and is its sole target in rishirilide gene cluster.

### 3.5. RslR4 Negatively Regulates Transcription of MFS Transporter Gene rslT4 and Its Own Expression

RslR4 is a member of the MarR family of transcriptional factors, often involved in regulating antibiotic resistance, stress responses, catabolism of aromatic compounds, and virulence regulation [40,41,42,43,44]. Most of the members are known to be transcriptional repressors of genes located next to it, including transporters of the major facilitator superfamily (MFS). Gene *rslR4* is located upstream and divergently oriented to *rslT4*, encoding MFS membrane protein. It is predicted to be an efflux pump providing strain self-resistance to rishirilide(s) and pathway intermediates. We assumed that RslR4 negatively controls *rslT4* transcription until rishirilide and/or its intermediate are accumulated.

To study the function of RslR4 on transcription of *rslT4* and its own gene, two plasmids pGUS-rslT4p and pGUS-rslR4p were constructed. In these plasmids, promoters of respective genes are fused with the β-glucuronidase reporter gene *gusA*. Constructs were introduced into S. *bottropensis* and into *S. bottropensis* ∆R4. Both strains carrying pGUS-rslT4p were grown in YMPGv medium. After two days, β-glucuronidase activity in the lysates was measured. In comparison to the wild type strain, *S. bottropensis* ∆R4 showed ca. 5-fold higher β-glucuronidase activity, indicating that RslR4 represses *rslT4* transcription (Figure 5a). The same experiment was carried out with the strains harboring pGUS-rslR4p. Here, a 2-fold increased β-glucuronidase activity in *S. bottropensis* ∆R4 in comparison to the wild type strain was observed (Figure 5b). This result concludes that RslR4 also negatively controls transcription of its own gene, acting as an autorepressor.

To confirm direct binding of RslR4 to intergenic region between *rslR4* and *rslT4*, a recombinant C-terminal histidine-tagged protein was heterologously produced in *E. coli* (DE3) Star, and its DNA-binding ability was analyzed by EMSA. Figure 5c shows a formation of retarded bands with increased concentration of RslR4. MarR regulators act as dimers and binds to palindromic sequences within target DNA [45]. The existence of several protein-DNA complexes indicates the presence of multiple binding sites in a given DNA sequence. Multiple sites could facilitate differential regulation of *rslT4* and *rslR4* genes. Next, a competition assay was carried out to prove binding specificity. As shown in Figure 5d, an excess of non-radioactive probe specifically competed with the labeled probe for binding to RslR4.

### 3.6. Rishirilides and Intermediates of the Pathway Interact with RslR4

The final product or pathway intermediate(s) often exert a regulatory role on strain resistance by modulating activity of cognate regulator. To assess the effect of end products and key intermediates on DNA binding by RslR4, the recombinant protein was incubated with the *rslR4-T4* region in reaction mixture containing different concentrations of putative ligands. In addition, we tested the newly isolated alkene derivatives RSH-O7a-d, which are accumulated in an enoyl reductase RslO7-deficient mutant. As shown in Figure 6, the interaction of RslR4 with its ligand molecules is concentration dependent. Rishirilides, RSH-O5, RSH-O9, RSH-O7a, and lupinacidin A at high concentration could eliminate DNA–protein interaction (Figure 6a–e,g). Little interaction was observed for RSH-O7b-d (Figure 6h,i) and no interaction was observed for the shunt product galvaquinone A (accumulated in ∆*rslO1* and ∆*rslO4* mutants) (Figure 6f). This result demonstrated that RslR4 DNA-binding ability is regulated by the final products and some key intermediates of the rishirilide pathway.

Altogether, these findings demonstrate that RslR4 negatively controls transcription of *rslR4* and *rslT4* genes by direct binding to their intergenic region. The repression is eliminated upon RslR4 interaction with cognate ligand molecules. This leads to the expression of the MFS transporter gene *rslT4* and provides strain with a resistance to synthesized metabolites.

## 4. Discussion

Streptomycetes are a rich source of natural products. Genes involved in the biosynthesis of secondary metabolite are joined together in the biosynthetic gene cluster, and usually encode one or a few regulatory proteins. Cluster-situated regulator(s) directly control transcription of biosynthetic genes, acting as activators or repressors. Moreover, expression of the regulatory genes is under control of different higher-level regulatory systems. Such cascade regulation coordinates production of secondary metabolite accordingly to strain’s physiology, developmental state, and environmental conditions.

Unlike many gene clusters that encode only one single pathway specific regulator, the *rsl*-gene cluster harbors four regulatory genes. Bioinformatic analysis of the proteins revealed three SARPs, RslR1, RslR2, and RslR3, and the MarR family transcriptional factor RslR4. SARP-like regulators usually activate a cognate biosynthetic gene cluster. They positively regulate transcription of biosynthetic genes and/or another SARP in the cluster.

In this study, we investigated the impact of four cluster-situated regulatory proteins on biosynthesis of rishirilides. Inactivation of each SARP completely abolished production of rishirilides and intermediates of the pathway, whereas deletion of *rslR4* revealed no distinct effect. Consistently, overexpression of *rslR1*, *rslR2*, or *rslR3* in *S. bottropensis* led to an increase of rishirilides production. These results strongly suggest that these transcriptional factors function as activators, all influencing rishirilide biosynthesis.

The presence of several SARPs in a gene cluster often goes along with a cascade of pathway-specific regulatory steps. This was shown for biosynthesis of tylosin, pristinamycin, and polymyxin [17,18,19,46,47]. Expression of *rslR1*, *rslR3*, and *rslR4* remained at the same level in the wild type and mutants lacking one of the regulatory genes, indicating that their transcription is independent from the *rsl*-regulatory genes. In contrast, expression profile of the *rslR2* gene in the wild type and in mutant strains revealed that its transcription is dependent on RslR1 and RslR3. The reduced expression of *rslR2* in *S. bottropensis* ∆R3 suggests that *rslR2* is activated by RslR3. In contrast, RslR1 likely exerts a negative control on *rslR2*, as *rslR2* transcription is increased in *S. bottropensis* ∆R1. These findings are additionally confirmed by results of in vivo transcriptional analysis. Moreover, we also revealed RslR2 to be a self-repressor. The self-modulating activity of RslR2 could be explained by analogy to JadR1, a cluster-situated regulator from jadomycin biosynthesis in *S. venezuelae*. Upon binding of jadomycin, JadR1 activates the biosynthetic gene cluster and represses transcription of its own gene [48]. Analogous, rishirilides, or intermediates of the pathway may serve as ligands for RslR2 in a dose-dependent manner. In general, well-tuned regulation of *rslR2* by RslR1 and RslR2 may help to maintain the biosynthesis of compounds at certain level, avoiding inhibition of strain growth. To further evaluate regulatory significance and investigate the exact role of RslR1 and RslR2, the recombinant proteins need to be produced and apply in experiments. Unfortunately, despite all our attempts, we failed to obtain these regulators in a soluble form.

RslR3 might be the highest-level cluster-situated regulator acting on rishirilides biosynthesis. Based on the protein structure, its activity could be modulated by ATPase domain or phosphorylation governed by particular kinase(s). ATPase activity of the structurally similar regulators PolY and AfsR triggers polymerization of the proteins and enhance their DNA binding activity [19,39,49]. Moreover, in vivo experiments revealed that endogenous ADP/ATP pool affected PolY activity, indicating a possible sensor ability of the ATPase domain and its regulatory function in modulating protein activity in response to current physiological conditions [19]. In addition, an activity of AfsR was shown to be regulated by multiple kinases [50].

Similar to PolY, RslR3 regulates transcription of another cluster-situated regulatory gene *rslR2*. The evidence that RslR3 did not bind to other promoter regions of *rsl*-transcriptional unit concludes that *rslR2p* is the only direct target of this regulator.

To avoid self-inhibition by produced compounds, *S. bottropensis* encodes the regulator RslR4 to control an expression of the cognate exporter gene *rslT4*. Typical for MarR-like regulators, *rslR4* is divergently oriented to its target *rslT4* that allows specific binding of RslR4 to intergenic region between the genes to repress transcription of both. Indeed, in vivo transcriptional levels of both genes were upregulated in *S. bottropensis* ∆R4 in comparison to the wild type strain. Negative autoregulation may prevent excessive accumulation of the regulator. This allows the keeping of narrow range of the protein concentration and results in more sensitive response to ligands.

RslR4 modulates the expression of *rslT4* in response to rishirilides and compounds from their biosynthetic pathway. The particularly interesting result is that both, the linear tricyclic intermediates (RSH-O5 and RSH-O9) and rishirilide-scaffold compounds interact with RslR4, and act as repression-relieving ligands. The ability to respond to pathway intermediates indicates that *S. bottropensis* cells initiate mechanism of self-immunity before the amount of synthesized end products reaches a critical level. This also suggests that intermediates are sufficient to trigger export and couple the mechanism of strain immunity to a biosynthetic pathway.

Based on these findings, we propose the following regulatory scheme (Figure 7). Rishirilides biosynthesis requires expression of three cluster-situated regulators, RslR1, RslR2, and RslR3 since the individual gene deletions completely abolished the rishirilides production or any known pathway intermediate(s). The biosynthesis is likely triggered by RslR3, which is present in the cell pool, upon signal-induced activation of the protein. Regulatory control exerted by RslR3 is mediated through activation of *rslR2* by direct binding to its promoter region. Most probably, *rsl*-genes are mainly regulated by the pathway specific regulator RslR2 together with RslR1. The transcription of *rslR2* is negatively regulated by RslR1 and an autorepression mechanism, while the expression of *rslR1* is likely activated by other regulatory elements of the cell. RslR4 is a repressor of the MFS transporter gene *rslT4* and its own gene transcription. The end products and pathway intermediates release repression by RslR4 that results in production of RslT4, providing cells with resistance to the synthesized compounds.

## Figures and Tables

**Figure 1 microorganisms-09-00374-f001:**
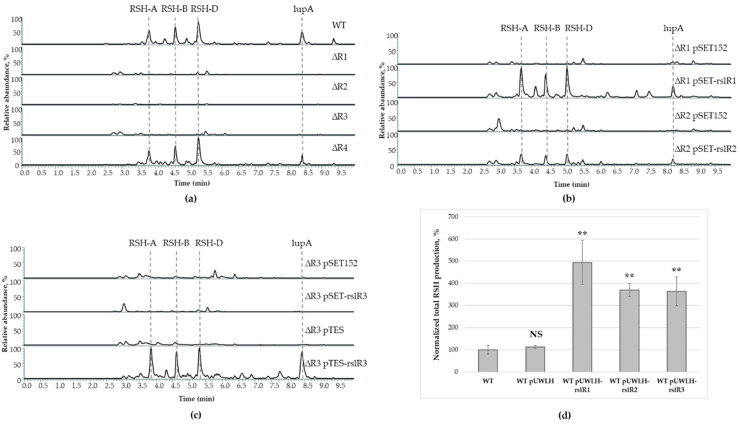
Rishirilides production by *S. bottropensis* strains. (**a**) UHPLC-MS analysis of culture extracts from *S. bottropensis* (WT) and mutant strains (*S. bottropensis* ∆R1, ∆R2, ∆R3, and ∆R4); (**b**) UHPLC-MS analysis of culture extracts from complemented *rslR1, rslR2* and (**c**) *rslR3* mutant strains. Strains carrying the empty vector pSET152 or pTES were used as a control. (**d**) Levels of total rishirilide (RSH) production by the wild type and strains which overexpress regulatory genes. The mean value of total rishirilide production in *S. bottropensis* (WT) and *S. bottropensis* pUWLH (WT pUWLH) was taken as 100%. Error bars represent standard deviations. Significant differences in total rishirilide production between *S. bottropensis* and strains overexpressing regulatory genes were calculated by a two-tailed *t*-test. Asterisks represent the significance value (** *p* < 0.01), whereas NS refers to not-significant differences.

**Figure 2 microorganisms-09-00374-f002:**
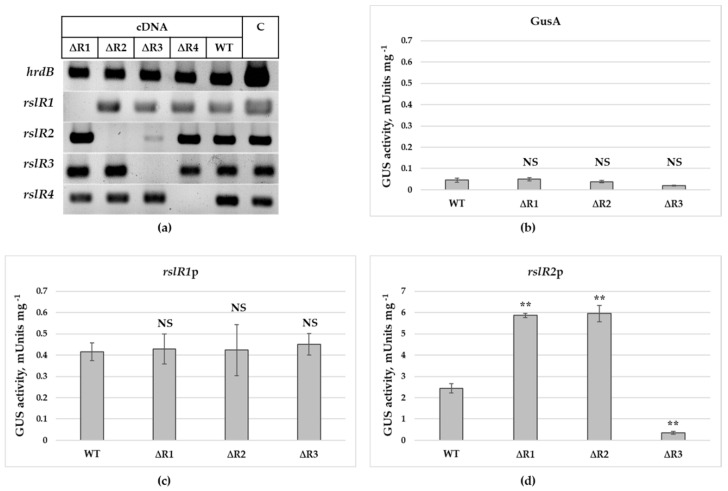
Transcriptional profile of regulatory genes in the wild type strain and in different mutants. (**a**) sqRT-PCR analysis of *rslR1*, *rslR2*, *rslR3*, *rslR4*, and *hrdB* transcripts in S. *bottropensis* strains. PCR was performed using cDNA and chromosomal DNA (C) as a template. β-glucuronidase activity from a promoterless construct (**b**), a construct containing the *rslR1* promoter (**c**), a construct containing the *rslR2* promoter (**d**) determined in the wild type strain and in S. *bottropensis* ΔR1, S. *bottropensis* ΔR2, and S. *bottropensis* ΔR3. Cultures and subsequent β-glucuronidase measurements were done in triplicate. Values were normalized to equal amounts of dry biomass. Error bars represent standard deviations. Significant differences in β-glucuronidase between the wild type and tested strains were calculated by a two-tailed *t*-test. Asterisks represent the significance value (** *p* < 0.01), whereas NS refers to not-significant differences.

**Figure 3 microorganisms-09-00374-f003:**
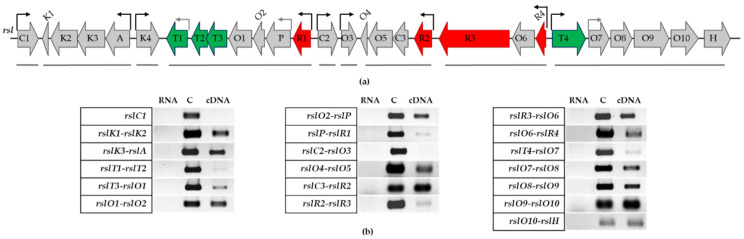
Transcriptional organization of *rsl*-genes. (**a**) Genetic organization of the rishirilide gene cluster. Regulatory, biosynthetic, and transporter genes are highlighted in red, grey, and green, respectively. The solid lines under the genes depict transcriptional units, black bent arrows are promoter regions, grey bent arrows are putative promoters; (**b**) RT-PCR analysis of *rsl*-genes co-transcription. RNA, genomic DNA (C), and cDNA were used as a template in PCR reactions. The functions of *rsl*-genes are described in [30,33,35].

**Figure 4 microorganisms-09-00374-f004:**
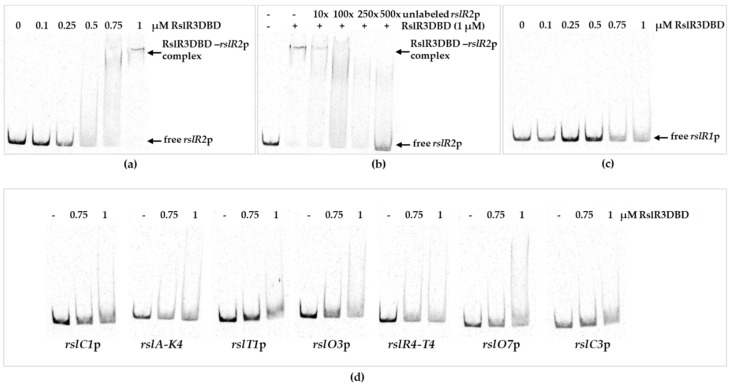
Electrophoretic mobility shift assay to study binding of RslR3DBD to promoter region of *rslR1*, *rslR2*, and structural genes. (**a**) EMSA containing *rslR2*p; (**b**) competition assay of RslR3DBD (1 µM) with *rslR2*p carrying 10, 100, 250, and 500-fold excess of unlabeled probe; (**c**) EMSA containing *rslR1*p; (**d**) EMSA containing promoters of structural genes.

**Figure 5 microorganisms-09-00374-f005:**
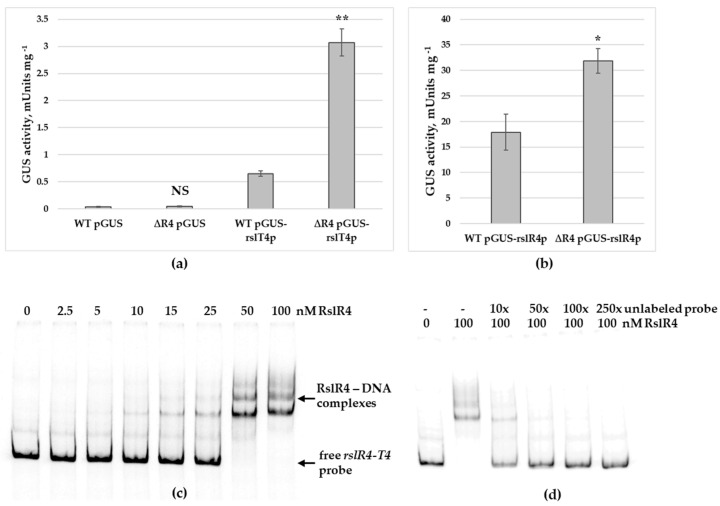
RslR4 negatively regulates transcription of *rslT4* and *rslR4* genes. (**a**) Transcriptional activity of *rslT4* promoter in *S. bottropensis* and in *S. bottropensis* ∆R4. (**b**) Transcriptional activity of *rslR4* promoter in *S. bottropensis* and in *S. bottropensis* ∆R4. Cultures and subsequent β-glucuronidase measurements were done in triplicate. Values were normalized to equal amounts of dry biomass. Error bars represent standard deviations. Significant differences in β-glucuronidase between *S. bottropensis* and *S. bottropensis* ∆R4 were calculated by a two-tailed *t*-test. Asterisks represent the significance value (* *p* < 0.05, ** *p* < 0.01), whereas NS refers to not-significant differences. (**c**) Electrophoretic mobility shift assay to study interaction of RslR4 with *rslR4-T4* region. Increasing concentrations of RslR4 were mixed with labeled *rslR4-T4* DNA probe. (**d**) Competition assay of RslR4 (100 nM) with *rslR4-T4* region carrying 10-, 50-, 100- and 250-fold excess of unlabeled probe.

**Figure 6 microorganisms-09-00374-f006:**
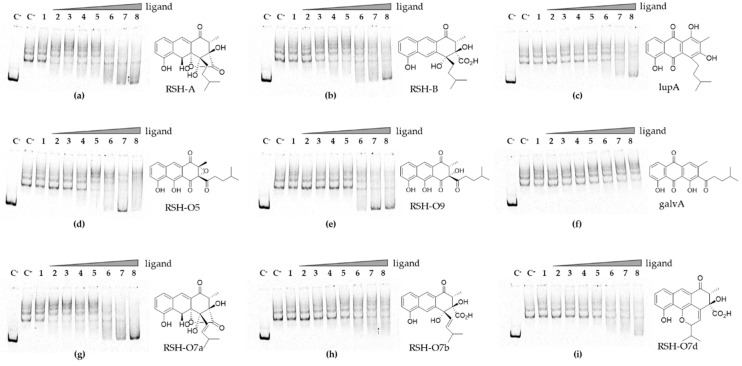
Electrophoretic mobility shift assay used to analyze effects of ligand molecules on RslR4 binding to the *rslR4-T4* region. RslR4 (100 nM) was incubated with the *rslR4-T4* region and increasing concentrations of the ligands (**a**) rishirilide A, (**b**) rishirilide B, (**c**) lupinacidin A, (**d**) pathway intermediate RSH-O5, (**e**) pathway intermediate RSH-O9, (**f**) shunt product galvaquinone A, (**g**) RSH-O7a, (**h**) RSH-O7b, (**i**) RSH-O7d. Lanes C^−^, negative control containing only labeled *rslR4-T4*, lanes C^+^, positive control containing RslR4 and labeled *rslR4-T4*, lanes 1, positive control containing DMSO, lanes 2–8, samples containing different concentrations of the ligands (1, 10, 50, 100 µM and 0.25, 0.5, 1 mM, respectively).

**Figure 7 microorganisms-09-00374-f007:**
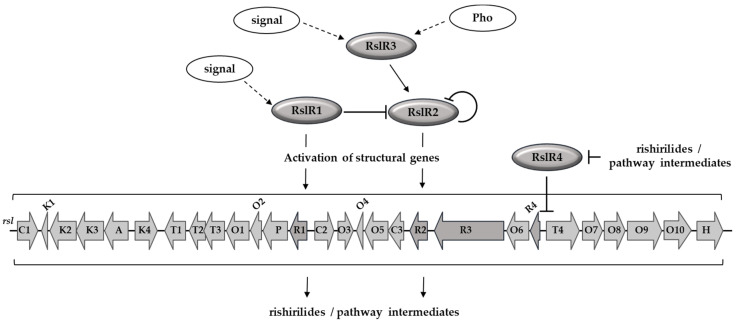
Proposed regulatory scheme of rishirilides biosynthesis.

## Data Availability

The data presented in this study are available in this article and Appendix A.

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
