# Peer review of "Regulatory Control of Rishirilide(s) Biosynthesis in Streptomyces bottropensis"

_microorganisms, 2021, doi:10.3390/microorganisms9020374_

Round 1

Reviewer 1 Report

Tsypik et al. studied the rishirilide biosynthetic gene cluster in Streptomyces bottropensis. Based on sequence similarity they identified four putative regulatory proteins and studied their function by gene knock-out, overexpression and in vitro studies on promoter binding.

Overall, the study was conducted very carefully, and the respective findings were cross validated with several methods. In terms of storytelling, I think the manuscript could still be improved (see below).

  1. The abstract is very dry and somewhat undersells the findings of the study. After reading it, I don’t really know why I should care about this study à what’s the knowledge gap / biological question you are addressing? It is also unclear where the introductory part ends and where the results summary starts. Please re-write.
  2. The cloning section of the methods (2.2 and 2.3) is not very detailed. I don’t think that I would be able to repeat your steps based on what is written. Please include more information on homology arms (length and nature), integration sites etc.
  3. Figure 1:
    • please arrange panels by occurrence in the text à panel B should become D;
    • please refer to specific panels throughout the text;
    • please add a sentence in the main text to describe why you used two constructs for the complementation of R3 and what the difference is between the two.
  4. Section 3.3: It is somewhat unclear to me why this section is here. To me it completely interferes with the main story. I would either move it to the beginning of the results section, which would also make it easier for the reader to visualize the complex gene cluster; or move it to the SI (not my preferred solution). I would also appreciate if the figure caption explained the abbreviations (“K”, “T”, “O”). I assume I know their meanings, but I can’t be sure.
  5. Figure 4: I find it very hard to grasp the content of the figure since the panels are hard to make out. Maybe an additional visual separation would be helpful (boxes/ lines)
  6. Figure 7: would it be worth adding a (putative) function to T4 here?

Minor:

Line 17: should be “overexpression”

Line 30 and all occurrences in introduction: “natural products” is a technical term and should be used rather than “natural compounds”.

Line: 73: please specify that you are only referring to overexpression of R1-R3 in this sentence

Line 83: “[…] and release the protein […]”

Line 259: italicize species names

Reviewer 2 Report

The manuscript „Regulatory Control of Rishrilide(s) Biosynthesis in Streptomyces bottropensis” investigate regulation of expression of the biosynthetic gene cluster responsible for production of rishrilide(s) secondary metabolites in S. bottropensis. The authors have generated 4 deletion mutants, one for each regulatory gene in the cluster and investigated production of rishrilides by the mutants. They have showed that regulators rslR1, rslR2 and rslR3 were required for the metabolite’s synthesis, while rsl4 deletion did not abolished production of rishilides. Introduction of deleted regulatory sequences into the mutants was able to restore expression of the cluster which was investigated by the authors using GUS assay. Which altogether suggested that the rslR1-3 proteins are positive regulators of the cluster expression. Moreover, authors have investigated interaction between each regulator using semi-quantitative RT-PCR and mobility shift assays. They have showed that rsl2 expression us regulated in positive manner by rslR3 and suppressed by rslR1. They have analysed organization of transcription unit in the cluster by RT-PCR with primers overlapping different genes in the cluster. The authors investigated which intermediates and products of rishilides synthesis pathways activate regulators proteins. Finally, the authors showed that rslR4 is negative regulation of rlsT4 and itself, which points to its role in resistance of S. bottropensis to produced rishrilides. All those results allowed authors to propose regulatory network of the cluster. The study provides important results on the regulation of the rishilides pathway. Examination and better understanding of the BGCs expression regulation is Streptomyces is crucial, as many of the clusters that could produce important from human point of view compounds are in dormant stage. Besides providing better understanding of the regulation of this particular pathway the research provides framework for studying regulation of other BGCs.

On the strong sites of the paper, the authors used number of molecular biology methods to elucidate regulatory network of the cluster. They have also investigated compounds that interact with the regulatory proteins even further describing the control of this cluster in S. bottropensis. The experiments were prepared thoughtfully, and presentation of results is clear and in logical order. I also appreciate detailed information of strain and primers used for the study as well as data on metabolites structures elucidation provided in supplementary materials.

However, there is few elements that could be improved and would require some additional explanation.

1) First, the description of statistical analysis for the GUS activity results is missing. There is information that the assay was performed in triplicates (were they biological or technical repeats) but besides this no additional information is provided. That is more striking taking into account that authors claim that the differences in activity they observed were or were not significant. Therefor, please provide information statistical method employed for the data analysis. Accordingly, charts on Figure 1, 2 and 5 should have significance indicators for the results that were significantly different as well as figure description should provide information of p-value. This would improve the presentation and scientific soundness of the manuscript.

2) Next thing is connected with the semi-quantitative RT-PCR on Figure 2. I am bit curious why the authors have used this approach instead of RT-qPCR which is more sensitive method that electrophoresis-based quantification. Also, it would be good if the authors provide whole gel pictures for each experimental setup (in supplementary materials) and not just cropped rows with bands of interest. Just for the clarity that amplification was specific.

3) Finally, in part of the paper the font should be changed to cursive when bottropensis is mentioned, this should be done particularly in line: 206, 227, 236, 243, 250, 259, 263, 264, 266, 267, 289, 293.

Yours sincerely,

Reviewer

Reviewer 3 Report

The article shows comprehensive approach to study rhisilirides pathway-specific regulation by 4 regulatory genes/proteins. I appreciate complex methodology approach, cross-verifications of results by multiple methods. The results are clearly presented and interpreted.

I have just minor comments:

1. I would appreciate some point added to the introduction concerning activity of rhisilides to inform the reader whether the interest on the cluster regulation is rather scientific and general or may have potential direct impact in future biotech applications.

2. In the methods section, concentration of chemicals in solutions, media etc. should be given without a space "0.5M NaCl" instead of "0.5 M NaCl".

3. Restriction endonucleases' names should have the first three letters in italics "EcoRV" instead of "EcoRV".
